# Role of IL-17A and IL-17RA in Prostate Cancer with Lymph Nodes Metastasis: Expression Patterns and Clinical Significance

**DOI:** 10.3390/cancers15184578

**Published:** 2023-09-15

**Authors:** Paweł Kiełb, Maciej Kaczorowski, Kamil Kowalczyk, Aleksandra Piotrowska, Łukasz Nowak, Wojciech Krajewski, Joanna Chorbińska, Krzysztof Dudek, Piotr Dzięgiel, Agnieszka Hałoń, Tomasz Szydełko, Bartosz Małkiewicz

**Affiliations:** 1University Center of Excellence in Urology, Department of Minimally Invasive and Robotic Urology, Wroclaw Medical University, 50-556 Wroclaw, Poland; kamil.kowalczyk@student.umw.edu.pl (K.K.); lukasz.nowak@student.umw.edu.pl (Ł.N.); wojciech.krajewski@umw.edu.pl (W.K.); joanna.chorbinska@student.umw.edu.pl (J.C.); tomasz.szydelko@umw.edu.pl (T.S.); 2Department of Clinical and Experimental Pathology, Wroclaw Medical University, 50-556 Wroclaw, Poland; maciej.kaczorowski@umw.edu.pl (M.K.); agnieszka.halon@umw.edu.pl (A.H.); 3Division of Histology and Embryology, Department of Human Morphology and Embryology, Wroclaw Medical University, 50-368 Wroclaw, Poland; aleksandra.piotrowska@umw.edu.pl (A.P.); piotr.dziegiel@umw.edu.pl (P.D.); 4Center for Statistical Analysis, Wroclaw Medical University, Marcinkowskiego 2-6, 50-368 Wroclaw, Poland; krzysztof.dudek@umw.edu.pl

**Keywords:** IL-17, IL-17A, IL-17RA, prostate cancer, lymph nodes metastases, radical prostatectomy

## Abstract

**Simple Summary:**

Prostate cancer (PCa) is the second most common type of cancer among men. The expression of IL-17A cytokine and its receptor IL-17RA may be used to predict the risk of aggressive prostate cancer. We examined the clinical data of 77 patients with PCa and lymph node metastasis (LN+) and then evaluated the levels of IL-17A and IL-17RA expression in the prostate and LN+. We found significant correlations between the investigated markers’ expression levels in examined tissues and clinical data, such as body mass index (BMI), the percentage of involved lymph nodes, or the European Association of Urology (EAU) risk group. The findings of this study suggest that IL-17A and IL-17RA may be useful in predicting the risk of aggressive prostate cancer; however, further studies are needed to determine their roles and potential clinical applications.

**Abstract:**

Prostate cancer (PCa) is the second most frequently diagnosed cancer among men. The use of IL-17A and its receptor IL-17RA as prognostic markers for PCa has shown promising results. We analyzed the clinical data of 77 patients with PCa after radical prostatectomy with lymphadenectomy and lymph node metastasis (LN+). We assessed the expression levels of IL-17A and IL-17RA in cancer cells in prostate and, for the first time, also in LN+. Prostate IL-17A expression positively correlated with BMI (*p* = 0.028). In LN+, the expression of IL-17A was positively correlated with the percentage of affected lymph nodes (*p* = 0.006) and EAU risk groups (*p* = 0.001). Additionally, in the group with high IL-17A expression in LN+, the extracapsular extension (ECE) of the prostate was significantly more frequent (*p* = 0.033). Also, significant correlations with the level of IL-17RA expression was found—expression was higher in prostate than in LN+ (*p* = 0.009); in LN+, expression positively correlated with the EAU risk group (*p* = 0.045), and in the group of high expression in LN+ ECE of lymph nodes was detected significantly more often (*p* = 0.009). Our findings support the potential role of IL-17A and IL-17RA as PCa markers; however, further studies are needed to determine their roles and potential clinical applications.

## 1. Introduction

Prostate cancer (PCa) is one of the major causes of cancer-related deaths in the male population and the second most frequently diagnosed cancer in men worldwide [1]. Owing to longer life expectancies and the fact that PCa incidence increases with patient age, there will be an increase in the number of PCa patients. The impact on society’s health will be even greater than it is now [2]. Despite the availability of therapy protocols that are constantly being improved, selecting the best course of action for a particular patient is challenging and the outcome is unpredictable. This is because more accurate tools are still lacking for determining survival prognosis and the likelihood of progression or metastasis following primary PCa treatment. Lymph node metastases are a significant risk factor for PCa patients and have a significant negative effect on survival and the risk of recurrence after primary treatment. Through the selection of appropriate adjuvant therapy and more stringent follow-up after primary therapy, nodal metastases also have an impact on the therapeutic process in patients [3,4]. Despite intensive improvements in technology, lymphadenectomy persists as a superior method in comparison to the use of radiological imaging techniques for the detection of positive (metastatic) lymph nodes (LN+) [5,6]. Radical prostatectomy (RP) with extended pelvic lymphadenectomy is the gold standard for identifying LN+. However, this is an additional challenging step added to an already complex operation, that is, RP. It is important to emphasize that lymphadenectomy does not improve survival and greatly increases the risk of side effects (e.g., longer hospital stay, increased blood loss, and a higher probability of lymphocele development) [7]. Extended pelvic lymphadenectomy should be performed in patients with intermediate- and high-risk PCa in the absence of more precise techniques to assess the lymph node status [8].

Inflammation is considered an increasingly important factor in the pathogenesis of many cancers, including PCa [9]. The inflammatory response is a complex process involving many different cells of the immune system and the chemokines and cytokines produced by them. Active oxygen and nitrogen radicals formed during inflammation are believed to be responsible for the suppression of antitumor activity and stimulation of carcinogenesis [10,11]. This inflammatory response probably promotes the survival, proliferation, and spread of tumor cells [12,13]. This is particularly important for the formation of PCa metastases. Many studies have shown a link between prostatitis and increased risk of developing PCa. This relationship has been observed in relation to chronic and acute prostatitis [14,15,16]. In addition, the occurrence of inflammation (mainly chronic) in patients with benign prostatic hyperplasia increases the risk of developing PCa, especially high-grade tumors [17]. Some authors also indicate that the evidence of the significant role of the inflammatory process in the development of PCa are studies that have shown that the use of antioxidants and anti-inflammatory drugs may reduce the risk of PCa [18,19].

Many different factors affect the treatment outcomes and prognosis of patients with PCa. This is due to the high heterogeneity of prostate tumors, which results in different treatment effects between patients. Currently, to determine the risk of progression, we rely on predictors such as the prostate-specific antigen (PSA) level or the stage and histological grade of PCa determined in the prostate biopsy material. In recent years, research on potentially new markers that may complement these known predictors has been gaining increasing interest. Preliminary conclusions from these analyses of the expression of immunohistochemical (IHC) markers in PCa, such as IL-17A and its receptor IL-17RA, suggest their potential usefulness in the process of improving diagnostics, determining the risk of progression (including metastasis), and response to primary and adjuvant treatment. It should be noted, however, that despite promising results, there are still too few unambiguous studies confirming the usefulness of these potential new PCa prognostic markers in clinical practice [20]. Therefore, routine assessment of their expression is currently not recommended by the urological guidelines.

One of the most important pro-inflammatory cytokines is IL-17. It is secreted by various immune cells, including helper T 17 cells and NK cells. Its precise effect on cancer pathogenesis is still not fully understood. According to the existing evidence, IL-17 has been suggested to promote angiogenesis, inhibit cancer cell apoptosis, and enhance cancer cell proliferation. Additionally, it has been hypothesized that IL-17 affects the development of a microenvironment favorable for cancer growth and potential metastasis [21]. Research has shown that it promotes the growth of colorectal, breast, pancreatic, and PCa cancers [22]. IL-17 is a cytokine family comprising six ligands (IL-17A–IL-17F) and five receptors (IL-17RA–IL-17RE) [23]. In this study, we examined IL-17A and its receptor IL-17RA. However, it is unclear how IL-17 contributes to PCa pathogenesis. Various studies have shown an increased expression of IL-17A and IL-17RA receptor in PCa and BPH cells [24,25,26]. According to previous studies, IL-17 has a stimulatory effect on PCa growth and metastasis even under castration conditions [27,28,29]. The results of various studies on the expression of individual ligands and IL-17 receptors in PCa remain unclear. For instance, in a relatively recent study, an increased expression of IL-17 was observed in low-grade PCa and BPH, whereas no expression of the IL-17RA receptor was detected in the tested material [30].

In this study, we extensively investigated the expression of IL-17A and IL-17RA in PCa cells from primary tumor tissues and LN+. Our study is unique because it is the first to analyze the expression of IL-17A and IL-17RA in LN+. To evaluate the utilization of the investigated markers as potential new negative risk factors for PCa progression, we compared the obtained results with the clinical data of patients with LN+.

## 2. Materials and Methods

### 2.1. Patients Selection

In this study, we included 77 patients with PCa who had lymph node metastases in the postoperative material. Between January 2012 and September 2018, all patients underwent RP with extended lymphadenectomy at the University Urology Center, Wroclaw, Poland. A retrospective clinical data analysis was performed on the study participants, and histopathological specimens collected during RP were selected for additional examination. An experienced uropathologist examined the selected specimens. The 2017 PCa Tumor, Node, Metastasis (TNM) classification and the Gleason system were used to evaluate tumor stage and grade. Additionally, classifications, including the European Association of Urology (EAU) risk categories for biochemical recurrence of localized and locally advanced PCa and the International Society of Urological Pathology (ISUP) 2014 grade (group) system, were used to better categorize patients. A PSA level 0.1 ng/mL at the first measurement after RP, typically six weeks after surgery, was used to determine the radicality of the procedure.

### 2.2. Tissue Microarrays (TMAs) and Immunohistochemical (IHC) Staining

For this study, we prepared histopathological samples for immunohistochemical staining and its further examination using the tissue microarrays (TMAs) technique. Sixteen TMAs were created for our study. The donor blocks were paraffin blocks containing material from the prostate with PCa or LN+. Donor blocks were then used to create histopathological slides stained with hematoxylin and eosin (HE). A Pannoramic Midi II histological scanner (3DHISTECH Ltd., Budapest, Hungary) was used to scan slides. Representative areas from the entire section were selected by a uropathologist using the Panoramic Viewer Program (3DHISTECH Ltd.). In order to further increase the representativeness of each case, 3 representative cores with a size of 1.5 mm from the donor block were chosen and transferred to the TMA’recipient’ block using the TMA Grand Master (3DHISTECH Ltd.). 

IHC reactions were performed on 4 μm TMA paraffin sections using an Autostainer Link48 (Dako, Glostrup, Denmark). Deparaffinization, rehydration, and antigen retrieval were performed using EnVision FLEX Target Retrieval Solution, High pH (97 °C, 20 min; pH 9), in PTLink (Dako). Endogenous peroxidase was blocked using the EnVision FLEX Peroxidase-Blocking Reagent (Dako) for 5 min. Primary antibodies—polyclonal rabbit anti-IL-17/IL-17A antibody (1:1600, cat. no NBP1-76337, Novus Biologicals, Minneapolis, MN, USA) and monoclonal mouse anti-IL17RA/IL-17R antibody (1:200, cat. No NBP2-25258, Novus Biologicals)—were applied for 20 min. Following this, the secondary antibody, conjugated with horseradish peroxidase (EnVision FLEX/HRP—20 min incubation), was applied. 3,3′-diaminobenzidine (DAB, Dako) was used as the peroxidase substrate, and the sections were incubated for 10 min. Finally, all sections were counterstained for 5 min with EnVision FLEX Hematoxylin (Dako). After dehydration in ethanol (70%, 96%, absolute) and xylene, all slides were closed with coverslips in SUB-X Mounting Medium in a coverslipper. The primary antibodies were diluted in the EnVision FLEX Antibody Diluent (Dako). The slides were scanned using a histologic scanner, Pannoramic MIDI (3DHistech). Reactions were evaluated with the use of Quant Center software (3DHistech) under researcher supervision. In order to evaluate the expression of IL-17A and IL-17RA, for every case, six TMA cores (3 from prostate and 3 from metastatic lymph node) were assessed using a Pannoramic Viewer Digital image analysis. 

Next, an experienced uropathologist who did not have access to patient clinical data assessed IL-17A and IL-17RA expression using the immunoreactive scale (IRS) developed by Remmele and Stegner [31,32] presented in Table 1.

The final IRS score was determined by multiplying the percentage of stained PCa cells (“A” score) with the staining intensity (“B” score). The prostate and LN+ samples from each patient were assessed independently. The final IRS score for the prostate and LN+ was calculated using the average score obtained from the assessment of each of the three cores of a specific tissue type. 

### 2.3. Statistical Analysis 

For quantitative variables, the mean, standard deviation (SD), minimum (Min), maximum (Max), median (Me), lower (Q1), and upper (Q3) quartiles were calculated. The empirical distribution of quantitative variables was examined to fit a normal distribution using the Kolmogorov–Smirnov and Shapiro–Wilk tests. Spearman’s rank correlation coefficient was calculated to assess the relationship between monotonic relationships between variables. Qualitative (nominal and categorical) variables were presented in contingency tables as numbers (n) and percentages (%). The significance of differences in quantitative parameters between the two groups was assessed using the Mann–Whitney U test, and the independence of the two qualitative factors was established using Pearson’s chi-squared test. In all analyzed cases, the associations were considered statistically significant at *p* < 0.05. Statistica v.13.3 (TIBCO Software Inc., Palo Alto, CA, USA) was used for all statistical analyses.

## 3. Results

The general characteristics of the patients are presented in Table 2. 

### 3.1. IL-17A

IL-17A expression in the prostate and LN+ was found in 98.7% (*n* = 76) and 100% (*n* = 77) of patients, respectively. Figure 1 shows a comparison of IL-17A expression levels in the prostate and LN+. 

As presented in Table 3, IL-17A expression levels were comparable in the prostate and LN+ (*p* = 0.415), with no statistically significant difference between the percentage of positively stained cancer cells (*p* = 0.634) and intensity of staining (*p* = 0.446).

A statistically significant positive correlation was observed between IL-17A expression levels in the prostate and LN+ (rho = 0.395; Figure 2a).

IL-17A expression (expressed by the IRS score) in the prostate and LN+ was not significantly correlated with patient age, postoperative GGG ISUP, or preoperative PSA level. IL-17A expression levels in the prostate and BMI were significantly positively correlated (*p* = 0.028). Additionally, a statistically significant positive connection between the level of IL-17A expression in LN+ and the percentage of the affected lymph nodes and the EAU risk group was found (*p* = 0.006 and *p* = 0.001, respectively). Table 4 presents the results of the statistical analyses. 

When analyzing the differences between the groups with low and high expressions of IL-17A (assessed based on the IRS score) and the pathological features or postoperative outcomes of patients, only one statistically significant correlation was detected between the extracapsular extension (ECE) of the prostate and the level of IL-17A expression in the LN+—it was significantly more common in the high expression group (*p* = 0.033). No statistically significant correlation was observed between these variables and IL-17A expression in the prostate (Table 5).

Next, the IRS scale variables—the percentage of IL-17A-positive cancer cells (“A” score in the IRS scale) and the intensity of staining IL-17A-positive cancer cells (“B” score in the IRS scale) in the prostate and LN+—were independently examined to further the analysis of the pathological characteristics or postoperative results of the patients. There was a correlation between the ECE of the lymph node and the percentage of IL-17A-positive cancer cells in the prostate (*p* = 0.009), as well as between the intensity of staining IL-17A-positive cancer cells in LN+ (*p* = 0.014).

### 3.2. IL-17RA

IL-17RA expression in the prostate and LN+ was found in 90.9% (n = 70) and 93.5% (n = 72) of patients, respectively. Figure 3 shows a comparison of IL-17RA expression levels in the prostate and LN+.

A statistically significant difference was observed in the level of IL-17RA expression between the prostate and LN+. The level of IL-17RA expression according to the IRS score was higher in the prostate than in LN+ (4 vs. 3; *p* = 0.009). In addition, the level of expression was significantly more often marked as low in the material from LN+ than in the prostate (90.3% vs. 74.3%; *p* = 0.012). IL-17RA, like IL-17A, showed a statistically significant positive correlation between expression in the prostate and expression in LN+ (rho = 0.369; Figure 2b). As shown in Table 4, there was only one statistically significant positive correlation between the EAU risk group and the level of IL-17RA expression (IRS score) in LN+. No significant correlations were observed between the level of expression in the prostate and the previously mentioned quantitative variables.

After analyzing the results to identify potential differences between the groups with low and high expression of IL-17RA (based on the IRS scale) and the pathological features or postoperative outcomes of patients were found only in the case of the frequency of ECE of the lymph node. This phenomenon was more common in the group with high expression of IL-17RA (71.4% vs. 20.0%, *p* = 0.009). Similar to IL-17A, no statistically significant correlation was observed between these variables and IL-17RA expression in the prostate (Table 6).

Furthermore, as with IL-17A, an in-depth analysis of the IRS variables (“A” score and “B” score) used to assess IL-17RA expression was performed. There were no statistically significant differences in this regard in either the prostate or LN+ samples.

## 4. Discussion

For clinicians and pathologists, PCa, the second most frequently diagnosed cancer in men, presents a significant diagnostic and therapeutic challenge. A rise in the number of new PCa diagnoses in men is anticipated in the near future. It is due to the correlation between PCa incidence and age and the rising life expectancy [2]. But, despite substantial progress in adjuvant therapy that have increased cancer-specific survival, we continue to base prognosis on conventional variables like PSA level, histological grade group, and clinical stage [33].

Numerous ongoing studies are investigating the function and use of IHC biomarkers in the diagnosis and prognosis of PCa, including the development of metastases. Although many of the findings from these studies are encouraging, urological guidelines for PCa currently do not take these findings into account [8,20]. 

IL-17A and IL-17RA are members of a large family of IL-17 cytokines that have been demonstrated to have both pro-cancer (in most cases) and cancer-inhibiting effects [20,34,35,36,37]. Zang suggested a mechanism of action for IL-17 in the development of PCa in a mouse model. According to the author, IL-17 promotes PCa carcinogenesis via matrix metalloproteinase 7 (MMP7), which is also increased in PCa and triggers epithelial-to-mesenchymal transition (EMT), resulting in the development of PCa [22].

Our research is novel because, despite the fact that IL-17A and IL-17RA expressions in the prostate have been assessed in a number of studies, no study has investigated this marker’s expression in LN+ [24,25,26,27,28,29,30]. 

The expressions of IL-17A and IL-17RA in the examined tissues can be regarded as an indicator of local inflammation, which is significant in the context of neoplastic processes. Studies have shown that a chronic inflammatory process can promote the occurrence and progression of neoplasms. The exact mechanism of action is not known, but one proposed explanation is that an inflammatory process can alter the tumor microenvironment, leading to the production of cytokines that promote tumor growth and metastasis. This is a component of the pre-metastatic niche theory, which proposes that the conditions for the formation of metastases are present when the microenvironment is favorable to cancer cells at the potential site of metastasis [38].

Studying the expression of inflammatory mediators in cancer cells is especially important because recent research has shown that the process of immune escape is one of the most important factors in the development of cancer. Cancer cells develop resistance to immune system neutralization as well as resistance to anticancer drugs during this process. Current research is focused on identifying factors that influence the development of immune escape cancer cells, as well as the development of effective anticancer immunotherapies [39,40]. Cytokines, including IL-17A, are an important group of factors that are involved in this process [41]. In a recent study, the authors showed that melittin, an anti-inflammatory drug, inhibits the proliferation and migration of castration-resistant prostate cancer cells by downregulating the IL-17 signaling pathway [42].

Available studies confirm that a chronic inflammatory process promotes the development of BPH and PCa by stimulating angiogenesis or stimulating cell growth, but so far, there is no clearly described mechanism of this action [43,44,45]. It has been shown that the infiltration of inflammatory cells selectively promotes the proliferation of prostate epithelial cells, which may be the source of PCa development [43]. In a study by De Marzo et al., the stimulating effect of the inflammatory process on the development of proliferative inflammatory atrophy (PIA) was found, which may be a precursor to the transformation into prostatic intraepithelial neoplasia (PIN) or PCa [44]. Despite these data, the effect of prostatitis on PCa progression has not been unequivocally demonstrated [46].

According to Liu et al., IL-17 stimulation increased the expression of proinflammatory genes, including IL-17RA, in mice, resulting in the development of a more aggressive form of PCa [47]. Another study by the same author found an increased expression of IL-17A and IL-17RA in PCa and BPH, concluding that IL-17A action through IL-17RA contributes to PCa development [26].

We found IL-17A and IL17RA expression in a very high percentage of prostate and LN+ samples (over 90% of all analyzed samples). In contrast to the findings of Janiczek et al., who did not find IL-17RA expression in either the prostate or BPH, our findings regarding IL-17RA expression in the prostate support the hypothesis made by Liu et al.

We discovered no significant differences in IL-17A expression between the prostate and LN+, either in terms of the percentage of positively stained cancer cells or the intensity of staining, which was high in both types of tissues in most cases. However, the level of expression of Il-17RA was significantly higher in the prostate than in LN+ (IRS score 4 vs. 3; *p* = 0.009). In general, the expression level of IL-17RA (assessed by IRS scale) was lower in both types of tissues examined than that of IL-17A, and the majority of samples showed a low level of expression (especially in the case of expression in LN +). Also, we observed that prostate expression of IL-17A and IL-17RA correlated positively with their expression in LN+. This suggests that IL-17A and IL-17RA are involved in metastasis formation and are a component of pre-metastatic niche formation in lymph nodes. Further research on PCa nodal metastases is needed to draw clear conclusions from these observations. Our findings need to be confirmed in future research. This could be a promising research direction for developing new systemic therapies for PCa, or it could be an additional factor influencing the estimation of the risk of nodal metastases.

We found no significant correlation between the level of expression of Il-17A and Il-17RA in the prostate and classic factors used to assess the risk of disease progression, such as PSA or EAU risk group. The only significant positive correlation found was between the level of IL-17A expression in the prostate and BMI (*p* = 0.028). This association could be related to the chronic inflammation seen in obese and overweight people, with increased expression being the result [48,49,50]. This is consistent with the findings of Liu’s studies on obese mice, in which he investigated the impact of hyperinsulinemia on the expression of IL-17 and its receptors as well as the progression of PCa [47].

When we evaluated the expression in the LN+, we observed a correlation between the expression levels of IL-17A (*p* = 0.001) and IL-17RA (*p* = 0.045) and the EAU risk group. This is the only significant association found between the classical model of PCa progression risk assessment and the expression levels of the markers investigated in this study. In addition, the level of IL-17A expression in LN+ was correlated with the percentage of affected lymph nodes (*p* = 0.006). These findings imply that IL-17A and IL-17RA may play a significant role in the development of pre-metastatic niches, although further evidence is required to support this theory. These results could eventually assist in improving the accuracy of models such as Memorial Sloan Kettering Cancer Center (MSKCC), Partin, and Briganti nomograms, which are used to assess the likelihood of PCa nodal metastases [51,52,53,54]. The continuous improvement of methods to assess the risk of the presence of nodal metastases is very important because, despite the currently used tools, approximately 70% of patients undergo unnecessary extended lymphadenectomy, showing the absence of nodal metastases [55]. It should be emphasized that lymphadenectomy is an additional element that increases the risk of complications and extends the duration of RP [7]. Research on new markers to increase the accuracy of risk assessment of lymph node involvement is extremely important. 

We found no statistically significant differences between groups with high and low IL-17A or IL-17RA expression in the prostate and clinicopathological characteristics of patients. In contrast, we observed these differences in LN+ expression. In the case of IL-17A, there was a significant difference in the frequency of prostate ECE; it occurred more frequently in the high-expression group than in the low-expression group. This finding is significant because the ECE of prostate is regarded as an independent risk factor for biochemical recurrence [56]. However, in the case of IL-17RA expression in LN+, a similar but not identical significant difference was noticed; the ECE of the lymph node was identified more often in the high expression group than in the low expression group. Furthermore, a statistically significant association was observed between the frequency of ECE in the lymph nodes and the percentage of IL-17A-positive cells in the prostate as well as the intensity of IL-17A staining in LN+. 

This study has some limitations. Firstly, there were no follow-up data for patients who underwent RP. The ability of this study to assess the correlation between IL-17A or IL-17RA expression and patient outcomes, such as biochemical recurrence or overall survival, is hampered by the absence of long-term data. Secondly, there are certain limitations to the IRS scale, which is the evaluation method used in this study to assess the expression of IL-17A and IL-17RA. A more detailed assessment can be performed using the H-score method [57,58], which requires more experience and time from a uropathologist. A simplified classification into groups of high and low expression may turn out to be too inaccurate when it comes to detecting subtle correlations. The method we used was a compromise between the accuracy of the analysis and available resources and research needs. Thirdly, we examined the expression of IL-17A and IL-17RA exclusively in PCa tissue without comparing them to the control group, such as lymph nodes from patients who had undergone RP and lymphadenectomy, and no LN+ was detected or tissues from benign prostatic hyperplasia obtained after the transurethral resection of the prostate (TURP). The last limitation of the study was that it involved a relatively small group of patients, which would have reduced its statistical ability to identify subtle differences and may have created bias.

The strengths of our work, derived from its novelty and the rigorous method we used, should not be diminished by the limitations we observed. Our work is unique in that it is the first to examine the expression of IL-17A and IL-17RA in PCa LN+; nevertheless, we see a need and plan to broaden our research in the future with a comparison to a control group, as mentioned above. This will further define the role of IL-17A and IL-17RA in PCa, as well as their potential clinical implications.

## 5. Conclusions

The results presented above show that IL-17A and IL17-RA have a statistically significant positive correlation between expression in the prostate and expression in metastatic lymph nodes. The prevalence of their expression suggests their role in local inflammation, which is associated with neoplastic processes. Our study is the first to assess IL-17A and IL17-RA expression not only in prostate tissue, but also in LN+. The findings of this study highlight the potential significance of IL-17A and IL-17RA in PCa metastasis and premetastatic niche formation. The correlations observed between marker expression and clinical parameters such as BMI and EAU risk point to possible links between chronic inflammation and disease progression. Although more evidence is needed, these markers could contribute to improved risk assessment models for nodal metastases, helping to avoid unnecessary lymphadenectomies. 

In summary, this study sheds light on the potential of IL-17A and IL-17RA as markers in PCa, and further studies, ideally with a control group and long-term outcomes, are required to determine the role and possible application of both markers in PCa.

## Figures and Tables

**Figure 1 cancers-15-04578-f001:**
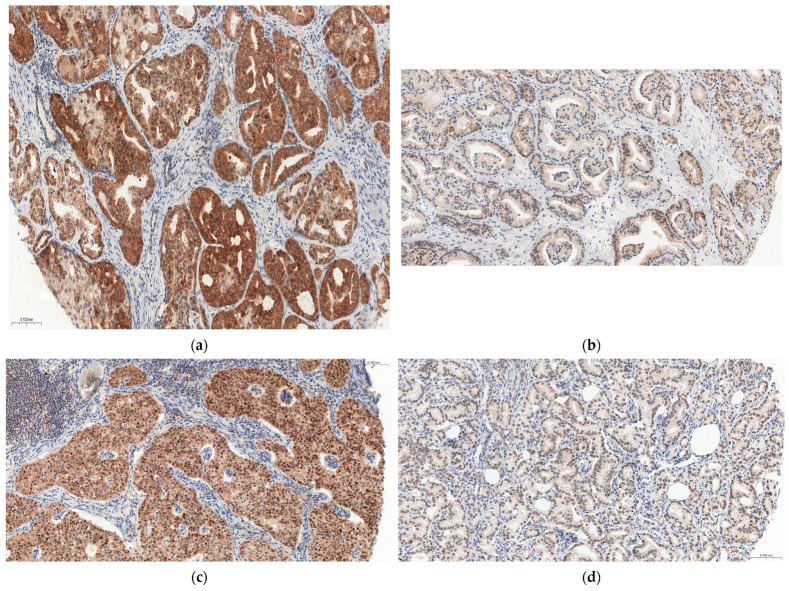
Comparison of IL-17A expression levels in the prostate and metastatic lymph nodes. (**a**) High IL-17A expression in prostate tissue; (**b**) Low IL-17A expression in prostate tissue; (**c**) High IL-17A expression in metastatic lymph node; (**d**) Low IL-17A expression in metastatic lymph node. Magnification, ×15.

**Figure 2 cancers-15-04578-f002:**
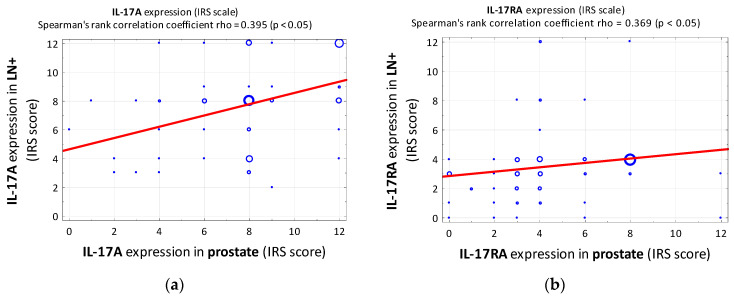
Summary of scatterplots and Spearman rank correlation coefficients. (**a**) Correlation between IL-17A expression in the metastatic lymph node and IL-17A expression in prostate (IRS score). (**b**) Correlation between IL-17A expression in the metastatic lymph node and IL-17A expression in prostate (IRS score). IRS—immunoreactive scale, LN+—metastatic/positive lymph node.

**Figure 3 cancers-15-04578-f003:**
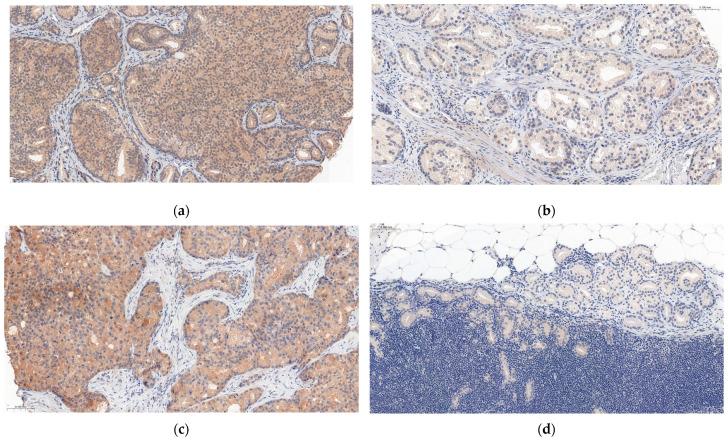
Comparison of IL-17RA expression levels in the prostate and metastatic lymph nodes. (**a**) High IL-17RA expression in prostate tissue; (**b**) Low IL-17RA expression in prostate tissue; (**c**) High IL-17RA expression in metastatic lymph node; (**d**) Low IL-17RA expression in metastatic lymph node. Magnification, ×15.

**Table 1 cancers-15-04578-t001:** Immunoreactive scale (IRS) by Remmele and Stegner. IRS score taking into account the percentage of positively stained prostate cancer cells (A) and the intensity of staining (B), and the final score is the result of multiplying these values (A × B). Based on the IRS score, patients were divided into groups of low and high IL-17A and IL-17RA expressions, respectively, as presented.

**Immunoreactive Scale (IRS)**			
**A—Percentage of Positive Cancer Cells**		**B—Staining Intensity**	
**Score**		**Score**	
**0**	no cells with positive reaction	**0**	no color reaction
**1**	<10% cells with positive reaction	**1**	mild reaction
**2**	10–50% cells with positive reaction	**2**	moderate reaction
**3**	51–80% cells with positive reaction	**3**	intense reaction
**4**	>80% cells with positive reaction		
**IRS SCORE (A X B): 0–12 points**			
**Final score**		**Level of expression**	
**1–7**		Low expression	
**8–12**		High expression	

**Table 2 cancers-15-04578-t002:** General characteristics and clinicopathological parameters of the patients. M—arithmetic mean, SD—standard deviation, BMI—body mass index, PSA—prostate-specific antigen, Me—median, Q1—lower quartile, Q3—upper quartile, EAU—European Association of Urology, n—number, %—percentage, pT—pathological tumor stage, GGG ISUP—International Society of Urological Pathology (ISUP) 2014 grade (group) system, radical procedure—defined as a PSA level <0.1 ng/mL at the first measurement after radical prostatectomy.

Variable	Statistics
**General characteristics of patients**	
Age (years):	
*M* ± *SD*	64.9 ± 5.5
BMI (kg/m^2^):	
*M* ± *SD*	28.1 ± 3.7
Preoperative PSA (ng/mL):	
*Me* (*Q*1; *Q*3)	19.8 (12; 36.1)
EAU risk group, *n* (%):	
Low-risk	1 (1.3)
Intermediate-risk	8 (10.4)
High-risk	38 (49.3)
High-risk locally advanced	30 (39)
**Clinicopathological parameters**	
pT, n (%):	
2a	1 (1.3)
2c	9 (11.7)
3a	14 (18.2)
3b	53 (68.8)
Postoperative Gleason, n (%):	
3 + 3	1 (1.3)
3 + 4	10 (13)
3 + 5	4 (5.2)
4 + 3	19 (24.7)
4 + 4	3 (3.9)
4 + 5	29 (37.6)
5 + 3	2 (2.6)
5 + 4	8 (10.4)
5 + 5	1 (1.3)
Postoperative GGG ISUP, n (%):	
1	1 (1.3)
2	10 (13)
3	19 (24.7)
4	9 (11.7)
5	38 (48.3)
Extracapsular extension of prostate, n (%):	
Yes	66 (85.7)
No	11 (14.3)
Extracapsular extension of lymph node, n (%):	
Yes	19 (24.7)
No	58 (75.3)
Resection margin, n (%):	
Positive	54 (70.1)
Negative	23 (29.9)
Neurovascular invasion, n (%):	
Yes	70 (90.9)
No	1 (1.3)
No data	6 (7.8)
Lymphovascular invasion, n (%):	
Yes	57 (74)
No	15 (19.5)
No data	5 (6.5)
Affected lymph nodes (%):	
*Me* (*Q*1; *Q*3)	12.5 (8.3; 27.3)
Radical procedure, n (%):	
Yes	36 (46.7)
No	41 (53.3)

**Table 3 cancers-15-04578-t003:** Basic descriptive statistics of the evaluation of IL-17A and IL-17RA expression in prostate and metastatic lymph node tissues and the results of comparisons. IRS—immunoreactive scale, A—percentage of positive cancer cells (value from IRS scale), B -staining intensity (value from IRS scale), Me—median, Q1—lower quartile, Q3—upper quartile, Min—minimum, Max—maximum, n—number, %—percentage.

	Expression (IRS Scale)					
	IL-17A			IL-17RA		
	Prostate	Metastatic Lymph Node	*p*-Value	Prostate	Metastatic Lymph Node	*p*-Value
A—Percentage of positively stained cancer cells (score)			0.634			0.271
*Me* (*Q*1; *Q*3)	4 [3; 4]	4 [3; 4]		3 [3; 4]	3 [2; 4]	
*Min*–*Max*	0–4	2–4		0–4	0–4	
B—Intensity of staining (score)			0.446			0.112
*Me* (*Q*1; *Q*3)	2 [2; 3]	2 [2; 3]		1 [1; 2]	1 [1; 1]	
*Min*–*Max*	0–3	1–3		0–3	0–3	
IRS score (A × B)			0.415			0.009
*Me* (*Q*1; *Q*3)	8 [6; 12]	8 [6; 9]		4 [3; 6]	3 [2; 4]	
*Min*–*Max*	0–12	2–12		0–12	0–12	
Expression level:			0.308			0.012
Low expression (1–7 score), n (%)	19 (25)	25 (32.5)		52 (74.3)	65 (90.3)	
High expression (8–12 score), n (%)	57 (75)	52 (67.5)		18 (25.7)	7 (9.7)	

**Table 4 cancers-15-04578-t004:** Correlation analysis between IL-17A and IL-17RA expression in prostate and metastatic lymph node assessed in IRS score and quantitative variables. BMI—body mass index, PSA—prostate-specific antigen, EAU—European Association of Urology, %—percentage, GGG ISUP—International Society of Urological Pathology (ISUP) 2014 grade (group) system.

	IL-17A				IL-17RA			
	Prostate		Metastatic Lymph Node		Prostate		Metastatic Lymph Node	
	rho	*p*	rho	*p*	rho	*p*	rho	*p*
Preoperative PSA (ng/mL)	0.033	0.778	0.027	0.813	0.057	0.623	0.100	0.387
Affected lymph nodes (%)	−0.007	0.952	0.312	0.006	0.100	0.385	0.144	0.211
Age (years)	−0.037	0.752	−0.037	0.749	0.047	0.682	0.019	0.873
BMI (kg/m^2^)	0.251	0.028	0.013	0.912	0.096	0.404	0.079	0.494
EAU risk group	0.159	0.168	0.376	0.001	0.051	0.660	0.229	0.045
Postoperative GGG ISUP	−0.020	0.862	0.146	0.205	−0.111	0.335	−0.023	0.841

**Table 5 cancers-15-04578-t005:** Number (percentage) of patients in groups differing in the level of IL-17A expression (based on IRS score) in the material from the prostate or metastatic lymph node, risk factors, and results of tests of independence. IRS—immunoreactive scale, n—number, %—percentage, pT—pathological tumor stage, ECE—extracapsular extension, NVI—neurovascular invasion, LVI—lymphovascular invasion, radical procedure—defined as a PSA level <0.1 ng/mL at the first measurement after radical prostatectomy.

IL-17A Expression Level (IRS Score-Based)							
Variables		Expression of IL-17A in PROSTATE			Expression of IL-17A in METASTATIC LYMPH NODE		
		Level of Expression		*p*-Value	Level of Expression		*p*-Value
		Low (N = 19)	High (N = 57)		Low (N = 25)	High (N = 52)	
		*n* (%)	*n* (%)		*n* (%)	*n* (%)	
pT	3a and 3b	15 (79.0%)	51 (89.5%)	0.257	20 (80.0%)	47 (90.4%)	0.279
	2a and 2c	4 (21.0%)	6 (10.5%)		5 (20.0%)	5 (9.6%)	
ECE of prostate	Yes	16 (84.2%)	49 (86.0%)	1.000	18 (72.0%)	48 (92.3%)	0.033
	No	3 (15.8%)	8 (14.0%)		7 (28.0%)	4 (7.7%)	
Resection margin	Positive	11 (57.9%)	42 (73.7%)	0.251	18 (72.0%)	36 (69.2%)	0.986
	Negative	8 (42.1%)	15 (26.3%)		7 (28.0%)	16 (30.8%)	
ECE of lymph node	Yes	6 (31.6%)	13 (22.8%)	0.543	3 (12.0%)	16 (30.8%)	0.094
	No	13 (68.4%)	44 (77.2%)		22 (88.0%)	36 (69.2%)	
NVI	Yes	16 (100.0%)	53 (98.2%)	1.000	20 (95.2%)	50 (100.0%)	0.296
	No	0 (0.0%)	1 (1.8%)		1 (4.8%)	0 (0.0%)	
LVI	Yes	12 (80.0%)	44 (78.6%)	1.000	16 (69.6%)	41 (83.7%)	0.288
	No	3 (20.0%)	12 (21.4%)		7 (30.4%)	8 (16.3%)	
Radical procedure	Yes	7 (43.8%)	28 (58.3%)	0.389	12 (57.1%)	24 (54.5%)	0.944
	No	9 (56.3%)	20 (41.7%)		9 (42.9%)	20 (45.5%)	
Expression of IL-17A in metastatic lymph node	Low	8 (42.1%)	16 (28.1%)	0.287	XX	XX	XX
	High	11 (57.9%)	41 (71.9%)		XX	XX	
Expression of IL-17A in prostate	Low	XX	XX	XX	8 (33.3%)	11 (21.2%)	0.178
	High	XX	XX		16 (66.7%)	41 (78.8%)	

**Table 6 cancers-15-04578-t006:** Number (percentage) of patients in groups differing in the level of IL-17RA expression (based on IRS score) in the material from the prostate or metastatic lymph node, risk factors, and results of tests of independence. IRS—immunoreactive scale, n—number, %—percentage, pT—pathological tumor stage, ECE—extracapsular extension, NVI—neurovascular invasion, LVI—lymphovascular invasion, radical procedure—defined as a PSA level <0.1 ng/mL at the first measurement after radical prostatectomy.

IL-17RA Expression Level (IRS Score-Based)							
Variables		Expression of IL-17RA in PROSTATE			Expression of IL-17RA in METASTATIC LYMPH NODE		
		Level of Expression		*p*-Value	Level of Expression		*p*-Value
		Low (N = 52)	High (N = 18)		Low (N = 65)	High (N = 7)	
		*n* (%)	*n* (%)		*n* (%)	*n* (%)	
pT	3a and 3b	45 (86.5%)	16 (88.9%)	1.000	57 (87.7%)	6 (85.7%)	1.000
	2a and 2c	7 (13.5%)	2 (11.1%)		8 (12.3%)	1 (14.3%)	
ECE of prostate	Yes	47 (90.4%)	14 (77.8%)	0.222	58 (89.2%)	5 (71.4%)	0.209
	No	5 (9.6%)	4 (22.2%)		7 (10.8%)	2 (28.6%)	
Resection margin	Positive	37 (71.2%)	13 (72.2%)	1.000	45 (69.2%)	6 (85.7%)	0.665
	Negative	15 (28.8%)	5 (27.8%)		20 (30.8%)	1 (14.3%)	
ECE of lymph node	Yes	13 (25.0%)	3 (16.7%)	0.745	13 (20.0%)	5 (71.4%)	0.009
	No	39 (75.0%)	15 (83.3%)		52 (80.0%)	2 (28.6%)	
NVI	Yes	45 (97.8%)	18 (100.0%)	1.000	59 (100.0%)	7 (100.0%)	1.000
	No	1 (2.2%)	0 (0.0%)		0 (0.0%)	0 (0.0%)	
LVI	Yes	40 (81.6%)	12 (70.6%)	0.491	48 (80.0%)	6 (85.7%)	1.000
	No	9 (18.4%)	5 (29.4%)		12 (20.0%)	1 (14.3%)	
Radical procedure	Yes	25 (58.1%)	8 (50.0%)	0.769	31 (57.4%)	2 (33.3%)	0.394
	No	18 (41.9%)	8 (50.0%)		23 (42.6%)	4 (66.7%)	
Expression of IL-17RA in metastatic lymph node	Low	43 (87.8%)	16 (94.1%)	0.667	XX	XX	XX
	High	6 (12.2%)	1 (5.9%)		XX	XX	
Expression of IL-17RA in prostate	Low	XX	XX	XX	43 (72.9%)	6 (85.7%)	0.667
	High	XX	XX		16 (27.1%)	1 (14.3%)	

## Data Availability

The data are available from the authors upon reasonable request.

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
