# Peer review of "Role of IL-17A and IL-17RA in Prostate Cancer with Lymph Nodes Metastasis: Expression Patterns and Clinical Significance"

_cancers, 2023, doi:10.3390/cancers15184578_

Round 1

Reviewer 1 Report

In this manuscript, the authors found that IL-17A and IL-17RA may be used to predict the risk of prostate cancer. The authors assessed the expression levels of IL-17A and IL-17RA in prostate and first time evaluated these two markers in metastatic lymph node. They show that there are significant correlations between IL-17A/IL-17RA expression levels and body mass index (BMI), percentage of involved lymph nodes, or the European Association of Urology. They propose that IL-17A and IL-17RA as PCa markers. In principle, the study addresses an interesting issue. I have several major concerns regarding this paper, as outlined below.

Major comments:

1.IL-17AIL-17/RA is generally expressed in prostate. And here the IL-17A/IL-17RA expression level in prostate cancer and LN+ is not correlated with pT-tumor stage and PSA concentration which is normally used to diagnostic of PC. So, the IL-17A/IL-17RA expression level used as an indicator is insufficient, especially no controls. There should be at least baseline of IL-17A/RA expression level.

2.Do authors investigate the IL-17A in blood and is there correlation between PSA and IL-17A? 

3. Is there any marker used for LN+? Do authors investigate  the relationship bewteen  general  diagnostic marker with IL-17A/IL-17RA?  

Mimor comments:

1.      The last two rows in table 5 and table 6, “Expression of IL-17RA in metastatic lymph node” and “Expression of IL-17RA in prostate”, is this reversed?

2.        The rule in Fig.1 and Fig. 3 is not obvious-please bolded.

Author Response

Thank you for taking the time to read our manuscript. We appreciate your feedback and value your opinion. We have carefully considered your points and would like to respond to each of them.

Major comments:

1.IL-17AIL-17/RA is generally expressed in prostate. And here the IL-17A/IL-17RA expression level in prostate cancer and LN+ is not correlated with pT-tumor stage and PSA concentration which is normally used to diagnostic of PC. So, the IL-17A/IL-17RA expression level used as an indicator is insufficient, especially no controls. There should be at least baseline of IL-17A/RA expression level.

The fact that we did not show a statistically significant correlation between the expression levels of IL-17A/IL-17RA and clinical data, such as PSA or T stage, does not mean that our study has a low value. This may also be due to the fact that we included patients with confirmed nodal metastases, i.e. initially in the advanced stage of aggressive prostate cancer. Our results may be an important point of reference for future studies investigating the role of IL-17A/IL-17RA in the pathogenesis of prostate cancer (especially in the metastatic stage), as well as in other malignancies.

As indicated in the discussion, one of the weaknesses of our work is the lack of a control group. In the future, we plan to extend the scope of this study by comparing the results of IL-17A and IL-17RA expression in tissue samples from the normal prostate and BPH. It is then technically possible to determine the base expression value.

As mentioned in the Discussion section, the expression evaluation method we applied was not ideal. The method we used was a compromise between the precision of the analysis, available resources, and research demands. A more extensive evaluation can be performed using the H-score method, which requires more experience and time from a uropathologist. A simplified classification into groups of high and low expression levels may turn out to be too inaccurate for detecting subtle correlations. Compared to other studies measuring IL-17A/IL-17RA expression, our study is one of the most complete; most previous studies utilized much simpler and more subjective assessments based solely on staining intensity (e.g., 0, 1+,2+,3+). Some published studies have been limited to a dichotomous assessment of the presence or absence of IL-17A/IL-17RA staining. We believe that our extensive study contributes significantly to the understanding of the role of IL-17A/IL-17RA expression in the progression of prostate cancer, particularly when evaluated in metastatic lymph nodes, which has never been examined before.

In addition, the expression of IL-17RA was not obvious, and in a recently published study, it was not detected in the prostate tissue (Janiczek et. al), so our results are an important contribution to the development of research on IL-17 expression. Our study is innovative, particularly in terms of assessing the level of expression in metastatic lymph nodes. Our results are the first, there is nothing to compare them to, and we cannot accept an arbitrary value of the baseline expression.

Of course, we agree that based on currently available research, we cannot treat IL-17A and IL-17RA as prostate cancer markers with proven clinical value.  To confirm their usefulness, the scope of the research should be extended to include the evaluation of expression in the above-mentioned control groups.

2.Do authors investigate the IL-17A in blood and is there correlation between PSA and IL-17A?

Our study was based on the retrospective results of patients who underwent radical prostatectomy, and IL-17A blood levels were not measured in these patients. Collecting these data would be particularly difficult considering that the determination of IL-17A concentration in the blood is not a standard test performed in patients before planned surgery and that the group of patients we selected were patients in the advanced stage of prostate cancer with very limited estimated survival. This is important because some of them were treated more than 10 years ago and, unfortunately, it may not be possible to measure IL-17A in the blood today (and possibly unreliable due to adjuvant treatment). However, carrying out such measurements is far beyond the scope of our research. Nevertheless, we think this is a very good idea, and we will consider it in the future when designing prospective studies. Thank you for this suggestion.

  1. Is there any marker used for LN+? Do authors investigate the relationship bewteen general  diagnostic marker with IL-17A/IL-17RA?   

Currently, there are no markers for detecting prostate cancer metastases in the lymph nodes. As indicated in the introduction, this is a very important clinical problem that we encounter in our daily work as urologists. Currently, in accordance with the guidelines of urological societies, to estimate the risk of lymph node involvement in the course of prostate cancer, we used the EAU classification (including T-stage, PSA, and Gleason score), according to which patients from the intermediate- and high-risk groups should undergo extended lymphadenectomy because of the high risk of metastasis nodes. At this point, we would like to emphasize that in our study, we showed a correlation between the EAU classification and the level of expression.

In LN+ patients, the expression of IL-17A and IL-17RA was positively correlated with the EAU risk groups (p=0.001 and p=0.045, respectively).

Other available tools are mathematical models that consider various clinicopathological data to estimate the risk of lymph node involvement (e.g., the Briganti nomogram or Partin tables). Despite the intensive development of imaging methods, none of them are perfect in terms of detecting nodal metastases. Currently, the gold standard for the assessment of lymph node involvement is an extended pelvic lymphadenectomy performed during radical prostatectomy. Considering all these issues, we believe that our study focusing on the problem of nodal metastases is very important and makes an important contribution to the understanding of the molecular processes of prostate cancer metastasis.

Mimor comments:

  1. The last two rows in table 5 and table 6, “Expression of IL-17RA in metastatic lymph node” and “Expression of IL-17RA in prostate”, is this reversed?

Thank you for bringing this bug to our attention group descriptions, which have been reversed. We rechecked the tables and corrected the captions and minor errors that occurred during the transfer and formatting of the tables (which did not affect the results).

  1. The rule in Fig.1 and Fig. 3 is not obvious-please bolded.

Thank you for your comments. The scale marked on the tissue photographs was applied automatically. Owing to the compression of the photos in order to place them in one file, it may be difficult to see; however, in the supplementary materials, we have included high-resolution photos where its visibility is very good. We believe that the quality of the photographs will be perfect in the final version of the manuscript. In addition, at the end of each caption under Figures, we have included information at what magnification the photo was taken ("Magnification, x15").

If you have any further comments or specific areas you would like us to focus on, please let us know. We value your expertise and are dedicated to delivering a high-quality document.

Thank you for your time and attention.

Sincerely,

Authors

Reviewer 2 Report

The authors evaluated the expression levels of IL-17A and IL-17RA in patients with prostate cancer deduced the correlation between these two cytokines and several clinical traits, and then proposed the insight of using these cytokines as marks to assess the lymph node status. Materials and methods including the analysis process have been described in detail and analysis results can support conclusions basically. 

One concern is that the authors declared this work is the first time analyzing the expression levels of IL-17A and IL-17RA during lymph node metastasis. Do you think the correlations shown in this manuscript are specific to prostate cancer? or only IL-17A and IL-17RA can be used as marks among their cytokines family?  

Author Response

Thank you for taking the time to read our manuscript. We appreciate your feedback and value your opinion. We have carefully considered your points and would like to respond to each of them.

One concern is that the authors declared this work is the first time analyzing the expression levels of IL-17A and IL-17RA during lymph node metastasis. Do you think the correlations shown in this manuscript are specific to prostate cancer? or only IL-17A and IL-17RA can be used as marks among their cytokines family?  

We believe that the observed correlations are not specific to prostate cancer. Previous studies have shown a correlation between IL-17A and IL-17RA expression in other malignancies, such as colorectal cancer (PMID:37211606). We believe that other ligands and receptors of IL-17 may also serve as potential markers of prostate cancer aggressiveness; for example, assessing the expression of IL-17F or the IL-17RC receptor provides promising results (DOI:10.1155/2020/4910595). However, further research is required to accurately assess the potential clinical use of these cytokines.

If you have any further comments or specific areas you would like us to focus on, please let us know. We value your expertise and are dedicated to delivering a high-quality document.

Thank you for your time and attention.

Sincerely,

Authors

Round 2

Reviewer 1 Report

Thank you for the opportunity to review this revised manuscript. The authors have revised the manuscript in accordance with the comments.